# Supplementing Low-Sodium Bicarbonate–Calcic (Lete)^®^ Water: Effects in Women on Bone and Systemic Metabolism

**DOI:** 10.3390/metabo13111109

**Published:** 2023-10-24

**Authors:** Carmen Marino, Imma Pagano, Giuseppe Castaldo, Manuela Grimaldi, Maria D’Elia, Angelo Santoro, Aurelio Conte, Paola Molettieri, Chiara Parisella, Michela Buonocore, Anna Maria D’Ursi, Luca Rastrelli

**Affiliations:** 1Department of Pharmacy, University of Salerno, Via Giovanni Paolo II, 132, 84084 Fisciano, Italy; cmarino@unisa.it (C.M.); ipagano@unisa.it (I.P.); magrimaldi@unisa.it (M.G.); mdelia@unisa.it (M.D.); asantoro@unisa.it (A.S.); 2Department of Pharmacy and Ph.D. Program in Drug Discovery and Development, University of Salerno, Via Giovanni Paolo II, 132, 84084 Fisciano, Italy; 3NutriKeto_LAB Unisa—“San Giuseppe Moscati” National Hospital (AORN), Contrada Amoretta, 83100 Avellino, Italy; giuseppecastaldo@yahoo.it (G.C.); aurconte@gmail.com (A.C.); paolamolettieri@gmail.com (P.M.); chiara.parisella@gmail.com (C.P.); 4National Biodiversity Future Center (NBFC), 90133 Palermo, Italy; 5Department of Pharmacy, Scuola di Specializzazione in Farmacia Ospedaliera, University of Salerno, Via Giovanni Paolo II, 132, 84084 Fisciano, Italy; 6Department of Chemical Sciences, University of Naples Federico II, Complesso Universitario di Monte Sant’Angelo, Via Cinthia 21, 80126 Naples, Italy; michela.buonocore@unina.it

**Keywords:** ^1^H-NMR, bicarbonate–calcium water, metabolomic, bone metabolism, osteoporosis

## Abstract

Calcium (Ca) represents about 40% of the total mineral mass, mainly in the bone, providing mechanical strength to the skeleton and teeth. An adequate Ca intake is necessary for bone growth and development in children and adolescents and for maintaining bone mineral loss in elderly age. Ca deficiency predisposes to osteopenia and osteoporosis. Healthy nutrition, including an adequate intake of Ca-rich food, is paramount to prevent and cure osteoporosis. Recently, several clinical studies have demonstrated that, in conditions of Ca dysmetabolism, Ca-rich mineral water is beneficial as a valuable source of Ca to be used as an alternative to caloric Ca-rich dairy products. Although promising, these data have been collected from small groups of participants. Moreover, they mainly regard the effect of Ca-rich mineral water on bone metabolism. In contrast, an investigation of the effect of Ca supplementation on systemic metabolism is needed to address the spreading of systemic metabolic dysfunction often associated with Ca dysmetabolism. In the present study, we analyzed urine and blood sera of 120 women in perimenopausal condition who were subjected for six months to 2l daily consumption of bicarbonate–calcium mineral water marketed under ^®^Lete. Remarkably, this water, in addition to being rich in calcium and bicarbonate, is also low in sodium. A complete set of laboratory tests was carried out to investigate whether the specific water composition was such to confirm the known therapeutic effects on bone metabolism. Second, but not least, urine and blood sera were analyzed using NMR-based metabolomic procedures to investigate, other than the action on Ca metabolism, potential system-wide metabolic effects. Our data show that Lete water is a valid supplement for compensating for Ca dysmetabolism and preserving bone health and integrity.

## 1. Introduction

Bone mass and density are determined by genetics, hormones, physical activity, and nutrition. While genetic factors are important in growth and peak bone development, an adequate intake of bone nutrients is the main factor for optimal genetic potential expression and bone maintenance during adulthood [1]. An optimal Calcium (Ca) intake is necessary for bone health at all stages of life: for bone growth and development in children and adolescents and for maintaining bone mineral loss in elderly age [2]. Accordingly, dietary requirements for Ca are determined by the need for bone development and maintenance: higher during childhood and adolescence, during pregnancy and lactation, and for the elderly. The recommended dietary allowance for Ca varies between 700 and 1200 mg/day throughout life, as stated both at the international level by the United States Department of Agriculture (USDA), and at the Italian level by the Reference Levels of Nutrients and Energy Intake for the Italian population (LARN) [3].

Ca represents about 40% of the total mineral mass, mainly in the bone where, combined with phosphorus in hydroxyapatite crystals Ca_10_(PO_4_)_6_(OH)_2_, it provides mechanical strength to the skeleton and teeth [4]. Ca is unique among nutrients in that the body’s reserve is also functional: increasing bone mass is linearly related to reducing fracture risk. The remaining roles of body Ca in metabolism serve as a signal for vital physiological processes, including vascular contraction, blood clotting, muscle contraction, and nerve transmission [5].

Most of the Ca, not stored in the bones, is present in the blood serum as an ion. The concentration of serum ionized Ca is regulated in healthy subjects by the action of calciotropic hormones: parathyroid hormone (PTH), 1,25-dihydroxyvitamin D [1,25(OH)2D], Fibroblast Growth Factor 23 (FGF23), and calcitonin A, the decrease of which in serum Ca concentration increases the secretion of PTH, favoring the release of Ca from the bone and the Ca tubular reabsorption, thus reducing urinary Ca excretion. At the same time, PTH stimulates the secretion of 1,25(OH)2D in the kidneys, which favors active Ca absorption in the gut. Increases in serum Ca are reversed by calcitonin, whose action is controlled by 1,25(OH)2D and PTH secretion. FGF23 controls the phosphate serum levels and, therefore, indirectly also calcemia [6,7].

Ca deficiency predisposes to pathologies such as osteopenia, osteoporosis, kidney stones, cancer, hypertension, obesity, and insulin resistance [8,9,10]. Alteration in the Ca metabolism is frequent in women after menopause, reflecting systemic metabolic changes which occur in response to a deficit in hormonal estrogen regulation. As a result, approximately 30% of all postmenopausal women suffer from osteopenia or osteoporosis, a pathological condition characterized by decreased bone tissue density [11]. As a manifestation of a systemic metabolic condition, osteoporosis is frequently associated with other pathological metabolic conditions such as hyperglycemia, high blood pressure, and high BMI. These symptoms, typical of the metabolic syndrome and results of a systemic inflammatory condition, may worsen osteoporosis prognosis and increase the risk of bone fracture [12].

The control of Ca metabolism for the diagnosis of osteoporosis is based on the measurement of the bone mineral density; however, blood and urine laboratory tests are necessary to have integrative information on the health status of the bone and the bioavailability of Ca in response to dietary or pharmacological treatments. Based on the players involved in Ca metabolism, the concentrations of the following serum and urinary metabolites are considered: calcium, phosphate, magnesium, total proteins, PTH, 1,25(OH)2D (vitamin D), bone alkaline phosphatase, and beta cross laps; moreover, a complete urine test of fasting morning and urinary Ca as well as phosphate excretion is conducted and evaluated by 24 h urine collection [13].

The first intervention to prevent and cure osteoporosis is based on a healthy lifestyle, including correct nutrition with an adequate Ca intake (1000–1500 mg/day), and regular physical activity [14]. Despite that, food consumption surveys indicate that the diet in the West is generally poor in products rich in Ca since milk and its derivatives are considered fat and hypercaloric foods. To reduce the trend of low Ca intake, leading health authorities, such as the World Health Organization (WHO), recommend mineral water rich in Ca as a good source of Ca, easily assimilated and an alternative to caloric dietary products [3,15,16,17].

Water is the main constituent of the human body and is involved in many bodily functions, including being the carrier of nutrients to reach biological fluids and the primary vehicle to eliminate waste and toxins [18]. According to the National Recommended Energy and Nutrient Intake Levels (LARN) for water, a 1.2 to 2.5 L daily intake is essential to maintain body water equilibrium. However, needs may vary among people, depending on age, physical activity, personal circumstances, and weather conditions. The USDA reports that food, beverages, and drinking water intake should represent 2.7 L per day in women aged 19–50 [19]. Natural mineral waters are classified with respect to the mineral content. In particular, the 2009/54/EC directive established that “Water with Calcium” is that which includes Ca content > 150 mg/L [20]. Several clinical studies have been carried out to evaluate the possibility of using Ca from mineral water as a dietary supplement for Ca deficiency [6,21,22,23]. By measuring Ca and Ca-related metabolite concentrations in blood and urine, Ca from mineral water is demonstrated to be comparable or higher than that of Ca taken with dairy products [23,24,25,26].

These studies, in support of Ca-rich mineral waters as a valid integration for Ca deficiency, are conducted on small groups of participants. Moreover, they are exclusively focused on the effect of Ca-rich mineral water on Ca absorption and bone metabolism [27,28,29]. On the other hand, several metabolic changes are demonstrated to be associated with Ca deficit and bone dysmetabolism; therefore, data regarding a potential effect of Ca-rich mineral water on systemic metabolism are missing. To bridge this gap, in the present study, we analyzed urine and blood sera of 120 women in perimenopausal condition who were subjected for six months to 2l daily consumption of Ca-rich mineral water marketed under ^®^Lete. Remarkably, this water, in addition to being rich in Ca and bicarbonate, is also poor in sodium.

A complete set of laboratory tests was carried out to investigate whether the specific composition of ^®^Lete was such to confirm the therapeutic effects on bone metabolism previously observed for other bicarbonate–calcium waters. Second, in this study, we aimed to investigate possible system-wide metabolic effects occurring due to taking a bicarbonate–calcium water poor in sodium, other than the effect on Ca metabolism. In addition to the laboratory tests, the urine and blood of the women recruited in this study were analyzed using NMR-based metabolomic procedures.

Metabolomics is the large-scale study of metabolites within biological fluids, describing a fingerprint of the metabolic conditions of an organism. NMR spectroscopy combined with high-resolution mass spectrometry (HRMS) is a robust technique for metabolomic analysis [30,31]. Our NMR-based metabolomic analysis allowed for the observation of the metabolic changes associated with Ca-rich water treatment, which include, but are not limited to, Ca metabolism. Our data show that water rich in Ca and poor in sodium is a valid supplement to compensate for Ca dysmetabolism and preserve bone health and integrity. The improvement of Ca homeostasis is associated with the compensation of biochemical parameters related to systemic dysmetabolism, thus suggesting a beneficial effect in controlling eventual systemic inflammation.

## 2. Materials and Methods

### 2.1. Participants

A total of 120 perimenopausal women were recruited at the Avellino (Italy) Azienda Ospedaliera “San Giuseppe Moscati” between June 2021 and April 2022. Their demographics are reported in Table 1.

### 2.2. Inclusion and Exclusion Criteria

The inclusion criteria for participants were female sex, aged 40–65, and 19–30 Kg/m^2^ body mass index (BMI). The exclusion criteria were the inclusion in other study protocols, pharmacological therapies in progress (estrogen-replacement hormone therapy, therapy for osteoporosis, corticosteroid therapy, insulin therapy), Ca supplements, presence of medium–severe renal and hepatic diseases, and documented severe osteoporosis (T-score MOC < −2.5).

### 2.3. Study Design

The women were randomly and in a blinded manner divided into the intervention group (79 participants) and the control group (41 participants). The institutional Ethics Committee CAMPANIA NORD for clinical trials and bio-medical research located at Azienda Ospedaliera “San Giuseppe Moscati,” Avellino, Italy, approved the study protocol (Prot. N.: NKCL2101; reg. N. CECN/1522), which followed the Declaration of Helsinki, according to the International Guidelines of Good Clinical Practice and the regulations of clinical trials. Informed written consent was obtained from participants after providing information about the nature, purpose, and procedures of the study (ClinicalTrials.gov Identifier: NCT05854342, University of Salerno Protocol Record CECN/1522).

### 2.4. Clinical Assessment and Intervention

All the participants were subjected to chemical and biochemical laboratory tests. Blood samples were analyzed in the clinical laboratory using automated analyzers and available commercial kits. Quantitative evaluation of the following clinical parameters was conducted for all participants at T0 and T6: glycemia (GLIC); creatinine (CREA); sodium (Na); calcium (Ca); alkaline phosphatase (ALP); cholesterol (COL); high-density lipoprotein (HDL); low-density lipoprotein (LDL); phosphorus (P); magnesium (Mg); urine creatinine (CREA U); urinary sodium (U Na); urinary calcium (U Ca); urinary phosphorus (U P); urinary magnesium (U Mg); creatinine clearance (CL CR); albumin (ALB); parathormone (PTH); osteocalcin (OSTEO); vitamin D (1,25(OH)2D); insulinemia (INSUL); pH; specific weight (SPEC W); bicarbonate (HCO_3_^−^); ionized calcium (Ca++). In addition, the BMI at t0 was calculated as indicated in the inclusion and exclusion criteria of this study.

### 2.5. Dietary Assessment and Intervention

The 121 participants were tested before (T0) and after six months (T6) of treatment. Firstly, each patient was given a food frequency questionnaire (FFQ) to assess Ca consumption [32]. The Ca intake for each food was calculated by multiplying the intake of mg of Ca/100 g of the product with the number of times it was taken per week. According to LARN recommendations, the daily Ca intake of a woman between 40 and 65 years is between 800 and 1200 mg/day [8]. The participants’ average Ca intake before treatment was 901.41 mg/day for the intervention group and 920.37 mg/day for the control group. After that, they were subjected to clinical nutritional evaluation. Nutrition experts evaluated the patient’s nutritional habits at T0. From the nutritional analysis of 120 participants, a Mediterranean diet emerged (Appendix A). After the initial assessment, experts provided dietary recommendations, suggesting the consumption of low-index glycemic carbohydrates, 1.0 g/pro kilo per day proteins, and a high content of unsaturated lipids. Deprivation of acidic foods and controlled Ca intake <700 mg/day was recommended. Then, they were subjected to clinical nutritional evaluation and provided with dietary recommendations, suggesting the consumption of low-index glycemic carbohydrates, 1.0 g/pro kilo per day proteins, and a high content of unsaturated lipids. Deprivation of acidic foods and controlled Ca intake <700 mg/day was recommended.

The intervention-group women were recommended to drink 2 L of bicarbonate–calcic mineral water daily, while the control-group women were recommended to drink 2 L of CO_2_-added oligomineral water. The water was suggested to be consumed as 500 mL with meals and the remaining 1500 mL during the day. The experimental water selected for this study is the natural bicarbonate–calcic sparkling mineral water Lete (Acqua Lete^®^; Società Generale delle Acque Minerali, Pratella, CE, Italy), shipped directly to the testing laboratory from its bottling facility. The chemical composition of the water is reported in Table 2.

### 2.6. Sample Pre-Treatment for NMR Analysis

Serum and urinary samples for metabolomic analysis were collected following the standard operating procedure (SOP) for metabolomic-grade serum samples [33]. First, 5 mm heavy-walled NMR tubes were prepared using 200 µL of serum sample added to 300 µL of buffer phosphate (0.075 M Na_2_HPO_4_∙7H_2_O, 4% NaN_3_ in H_2_O used as preservative and 3-(trimethylsilyl)-2,2,3,3-tetradeuteropropionic acid, sodium salt (TSP-d4) used as an internal reference for the alignment and quantification of NMR spectra signals) at pH 7.4. A total of 1.5 mL of 24 h urine was previously centrifuged at 15.000× *g* for 10 min at 4 °C to pellet any particulates in the sample, and then the urine was filtered using a 0.2 µm filter. A total of 500 μL of urine was transferred into a new tube and mixed with 50 μL of 50 mM phosphate buffer in 99.8% D_2_O [33,34].

### 2.7. NMR Data Acquisition and Processing

NMR experiments were conducted at 298 K on a Bruker Avance 600 MHz spectrometer equipped with a 5 mm triple-resonance z-gradient. NMR spectra were processed and visualized using Topspin 3.2 (Bruker Biospin Bruker Biospin, Fällanden, Switzerland). 1D-NOESY experiments were conducted using the excitation sculpting pulse sequence for water suppression. The experiments’ results were collected using a 14 ppm sweep width, 192 transients of 16 complex points, and an acquisition time of 4 s transient and 60 ms mixing time. For identifying and quantifying metabolites, we used the online software Bayesil, (http://bayesil.ca, accessed on 1 September 2023) [35], which automatically processes and analyzes monodimensional ^1^H-NMR spectra of ultra-filtered biological samples. The glucose duplet signal at 5.4 ppm was used as a chemical shift reference, whereas the quantification was based on the peak intensity of TSP-d4, used as an internal reference compound.

### 2.8. Statistical Analysis

Biochemical parameters were analyzed using R-package 4.2.2 applying *t*-test and considering the parameters with *p*-value < 0.05 as significant [36]. The samples’ metabolomics data were normalized using sum, log-transformed, and Pareto-scaled and analyzed using MixOmics R-package (mixOmics-package) by the multivariate unsupervised method: principal component analysis (PCA) and multivariate supervised partial least-squares discriminant analysis (PLS-DA) [37]. A sample plot was used to observe the clustering of metabolomic profiles of the samples. In the graph, samples are represented as points positioned based on their projection on the selected latent components of the data. The correlations between the variables were plotted using a circular correlation plot obtained with the R Mix Omics package [37]. Data clustering was confirmed by the approach of distance metrics calculated by the centroid method, maximum distance, and Mahalanobis distance (Appendix A). The contribution of each variable for each component is represented in the loading plot, where the bar length corresponds to the importance of the variable in the construction of the given component, which can be positive or negative [37]. The analysis of the pathways was carried out by analysis with the Enrichment tool. The output of Metaboanalyst 5.0 was examined, and the KEGG paths [38] were chosen according to the lower false discoveries (FDR), *p*-value < 0.05, and the hits value related to the number of metabolites belonging to the pathway, >1. For a more precise visualization of common metabolites between the two biofluids, a Venn diagram was constructed using Interact Venn www.interactivenn.net (accessed on 20 August 2023) [39]. OmicsNet https://www.omicsnet.ca, accessed on 1 September 2023, was used for the creation of biological network and pathway prediction. The matrices of serum and urinary metabolites’ concentration, expressed by KEGG code, and the relative fold change value were analyzed by Omics-Net. The prediction of the pathways involved in the metabolomic profile was made by setting the Function Explorer analysis using the KEGG database [40].

## 3. Results

### 3.1. Clinical Analysis

The participants’ average Ca intake before treatment was 901.41 mg/day for the intervention group and 920.37 mg/day for the control group.

Clinical data deriving from blood and urine laboratory tests for 120 perimenopause women belonging to the intervention and control groups before (T0) (Appendix A) and after 6 months (T6) of water diet–therapy treatment were statistically analyzed using *t*-test to assess their significance (*p*-value). The analysis reporting clinical parameters indicates an increase in the urinary excretion of Na and Ca. Moreover, increased 1,25(OH)2D, Ca, and HCO_3_^−^ blood values were detected for most subjects in the intervention group at T6. On the contrary, a decrease in OSTEO and urinary-specific weight (SPEC. W) was observed (Figure 1, Appendix A).

The data regarding the laboratory tests for the patients in the control group show the absence of any significant change in the urinary and blood parameters at T6 (Appendix A). Interestingly, by comparing serum and urinary laboratory test data for the intervention and control group at T6, significant increases in the concentrations were observed for PTH, ALP, CREA U, U Ca, U Na, U P, U Mg, and Ca^2+^ in the intervention group at T6. On the other hand, OSTEO and glycemia decreased (Figure 2, Appendix A).

### 3.2. Multivariate Statistical and Pathway Analysis

Data matrices, including the urine and serum metabolites’ concentrations derived from NMR analysis, were analyzed using a multivariate PCA unsupervised approach and PLS-DA supervised approach (R package: MixOmics [37]) [41]. The matrices analyzed consisted of 73 metabolites detected in the urinary extract and 43 in the blood sera.

To identify possible modifications in the metabolomic profiles in response to the different water intakes for the intervention and the control group, we performed PLS-DA on the data matrices using serum and urine metabolites detected at T6 as variables (Figure 3). PLS-DA score plot identified a separation of intervention and control group serum (Figure 3A) and urine (Figure 3B) metabolomic profile at T6.

The benefit of Ca–bicarbonate water compared to CO_2_-supplemented oligomineral water is confirmed by the absence of clustering in the control group’s metabolomic profiles (Figure 4), in contrast to what was observed in the intervention group (Figure 5A,B).

To investigate which metabolites are primarily responsible for the differences in the intervention group’s metabolomic profiles at T0 and T6, we used *t*-tests with a *p*-value threshold < 0.05, and fold change with a threshold equal to or superior to 75% (Appendix A). In Figure 5B, the correlation circle plot applied to Components 1 and 2 indicates the variables that most affect the clustering (in the graph, the metabolites distant from the axes’ origin) and the correlation between the metabolites (evident in the proximity between the metabolites). Accordingly, an inspection of Figure 6B indicates that the metabolites significantly responsible for clusterization are isoleucine, succinate, and ketone bodies; on the other hand, a correlation exists between ketone bodies (3-hydroxybutyrate, 2-hydroxybutyrate and acetone) and succinate, as well as between isoleucine arginine and choline, as shown from the proximity of position of these metabolites on the plot.

Similarly to the serum extract, PLS-DA conducted on the urinary metabolites’ concentrations matrix revealed a different metabolomic profile of the intervention group at T0 and T6 (Figure 6A). The correlation circle plot highlighted that fatty acids such as 2-hydroxy isovalerate and 3-hydroxy isovalerate are responsible for the metabolomic modification between T0 and T6, and the changes of these metabolites correlate with changes in the concentrations of butyrate, tryptophan, threonine, dimethylamine, and hypoxanthine (Figure 6B).

Figure 7A shows the incidence of each metabolite for separating the blood sera metabolomic profiles of the intervention group at T0 and T6. Higher concentrations of ketone bodies (3-hydroxybutyrate, 2-hydroxybutyrate, and acetone) are evident for the intervention group at T6. Moreover, significant modifications in the amino acid concentrations result in increased Isoleucine and Ornithine and lower Arginine, Valine, and Phenylalanine concentrations.

A similar analysis for the urine metabolomic profile (Figure 7B) indicates a reduction in glucose, lactic acid, and pyruvate concentrations (suggestive of altered energy metabolism) and an increase in fucose, 3-hydroxyisovalerate, and 2-hydroxyisovalerate (Figure 7B); in addition, lower valine and tyrosine concentration, as well as higher lysine, leucine, and serine concentrations are found.

Interestingly, increased urinary succinate and isoleucine concentrations are associated with decreased serum succinate and isoleucine concentrations.

In order to understand which variables were mostly affected by Ca–bicarbonate water intake, we carried out the loading plot analysis using the serum and urinary matrices of the control and the intervention group at T6.

Figure 8A confirms the high concentrations of 3-hydroxybutyrate, succinate, and acetone in the intervention compared to the control group at T6. The impact of bicarbonate–Ca water intake on amino acid metabolism is confirmed by increased serum concentrations of isoleucine and proline and decreased urinary excretion of glycine, histidine, and glutamate in the intervention group at T6 (Figure 8B). To interpret the change in metabolite concentrations in terms of the modifications of biochemical pathways, we carried out pathway analysis using MetaboAnalyst 5.0 (Table 3). Furthermore, we identified the common dysregulated pathways between the two biofluids using the Venn diagram (Figure 9) [42].

Table 3 shows that, according to the change in metabolite concentrations, biochemical pathways related to fatty acids’ biosynthesis and ketone body metabolism are modified (pathways’ selection criteria are hits > 2 and *p*-value < 0.05). Moreover, those biochemical pathways related to amino acid metabolism appear significantly altered: arginine and proline; glycine and serine; glutamate, alanine, and methionine; valine, leucine, and isoleucine degradation. Finally, metabolite-altered concentrations in urine are consistent with the alteration in biochemical pathways related to galactose metabolism, fructose, and mannose degradation.

## 4. Discussion

Osteoporosis is a chronic disease characterized by a reduction in bone density. It is prevalent in postmenopausal women [43], causing a significant risk of bone fractures [44]. The primary intervention to control osteoporosis is a healthy lifestyle based on correct nutrition and regular physical activity. A diet with adequate Ca intake is a fundamental recommendation [45].

Milk and its derivatives are considered a primary Ca source [23,24,25,26]. However, their consumption has progressively been reduced as they are regarded as hypercaloric food. To favor the consumption of Ca in the diet, the leading health authority, WHO, has recently recommended diet supplementation with Ca-rich water. Indeed, in vitro and in vivo experiments have proved the efficacy of alkaline Ca-rich water to favor bone matrix affixing [13]. At the same time, several clinical studies have demonstrated that bicarbonate–calcic water is an efficacious nutritional supplement to preserve an adequate Ca content in pathological conditions characterized by bone matrix loss [46,47]. While all these data demonstrate the efficacy of bicarbonate–calcic water on Ca metabolism and bone status [6,13,48], data regarding the impact of bicarbonate–calcium water supplements on the systemic metabolic balance are missing.

Here, we reported an analysis of the blood and urine of 121 perimenopausal women using standard laboratory tests (Appendix A, Figure 1 and Figure 2) and NMR-based metabolomics analysis. The women were divided into an intervention group advised to drink 2l of bicarbonate–calcic water daily for 6 months, and a control group advised to drink 2l of CO_2_-supplemented oligomineral water daily.

Ca metabolism and bone health status may be appreciated by measuring blood Ca concentration and the markers responsible for its regulation: PTH, 25(OH)vitamin D, and calcitonin, as well as several ions and enzymes whose concentrations are linked to Ca, such as phosphate, magnesium, total proteins, and bone alkaline phosphatase [13].

Accordingly, all these parameters were monitored in the blood and by 24 h urine collection. After six months, the subjects drinking bicarbonate–calcium water showed increased blood and urine Ca concentration values, indicating a correct Ca absorption in response to the Ca-rich water treatment. In line with this metabolic change, increases in PTH, 1,25(OH)2D, and urinary excretion of P and Mg were evident; moreover, there was a decrease in OSTEO and urinary-specific weight. Interestingly, the latter are products of bone catabolism, suggesting that their decrease is a limitation of bone catabolism. In response to the Lete^®^ water treatment, the intervention group also exhibited an increase in urinary Na, typical of the diuretic water effect. Remarkably, this is specific, as it was not observed at T6 in the control group, although subjected to the 2l water treatment.

Clinical laboratory tests indicate that several parameters not strictly related to the Ca metabolism changed in the intervention group at T6 (Figure 2). For example, a decrease in glycemia related to energetic metabolism is evident, hinting at a more general effect on metabolism associated with the improved metabolism of Ca.

To explore this further, we performed an NMR-based metabolomic analysis of the urine and sera of the women enrolled in this study at T0 and T6. While the metabolomic profile of the women in the intervention and the control group at T0 is very similar, separation is observable at T6 (Figure 3A,B). This is due to the different effect of the Lete^®^ diet–therapy treatment, which induces significant metabolomic changes in the control group at T6, while the same is not observable for CO_2_-enriched mineral water (Figure 4A,B).

In search of the metabolites responsible for the modification of the metabolome in the intervention group, we performed loading plot analysis. Accordingly, we observed a statistically significant decrease in glucose (Figure 7B, Appendix A) after the intake of Lete. Moreover, the hypoglycemic effect is associated with modification of the succinate concentrations (increased in the serum and decreased in the urine at T6) (Figure 7A,B). By looking at the biochemical pathway to comprehend the significance of these metabolic changes, we observed that the hypoglycemic effect associated with the modification of the succinate concentrations may reflect an enhancement of the mitochondrial activity (Table 3, Figure 9). A significant amount of scientific data shows that altered mitochondrial activity induces systemic metabolic dysfunction and inflammation.

These data show that bicarbonate–calcium water treatment, in addition to the effects on Ca and bone metabolism, may improve systemic inflammatory conditions. Interestingly, severe osteoporosis, with a high risk of bone fracture, is frequently associated with several symptoms of metabolic syndrome and hyperglycemia, suggesting that a general condition of energy dysmetabolism and systemic inflammation also worsens this pathology [49,50].

The pathway analysis confirms these data, indicating that the changes in the metabolite concentrations are associated with the modulation of metabolic biochemical networks related to the improvements in the mitochondrial activity’s (i) citric acid cycle, (ii) gluconeogenesis, (iii) acyl group transfer, and (iv) pyruvate metabolism (Table 3, Figure 9).

The metabolomic profile of women in the intervention group is characterized by increased concentrations of several metabolites associated with the stimulation of tissue repair, bone fixing, and collagen synthesis. Specifically, we observed high concentrations of ornithine, an amino acid involved in collagen synthesis (Figure 8A) [51]. Stimulation of collagen synthesis was also confirmed by the upregulation of arginine and proline metabolism, evident from the blood and urine pathway analysis (Figure 9 Table 3). Then, we observed high concentrations of succinate, 3-hydroxybutyrate, and acetone, metabolites involved in the in vitro differentiation of murine osteoblast and in vivo bone mineralization (Figure 8A,B) [52,53]. In addition, previous scientific studies have shown that the variation in the ketone bodies, in particular of 3-hydroxybutyrate and the energy pathways shown in Table 3, could predict increased osteoblast activity against osteoclast remodeling [50,53,54]

According to sera and urine metabolomic data, mineral water treatment induces modifications in the biochemical pathways related to the metabolisms of the branched-chain amino acids valine, leucine, and isoleucine, and aromatic amino acid phenylalanine and arginine (Table 3). This confirms previously published data according to which high valine, arginine, and phenylalanine concentrations may be osteoporosis biomarkers [55].

Taken together, our data show that bicarbonate–calcic water intake for six months, in addition to inducing an improvement in the biochemical parameters related to Ca metabolism and bone health, can (i) restore mitochondrial energy pathways, (ii) promote the application of bone mass and collagen, and (iii) reduce the biomarkers of osteoporosis. Therefore, the alkaline environment improves bone metabolism toward a lower risk of osteoporosis and bone fractures.

## Figures and Tables

**Figure 1 metabolites-13-01109-f001:**
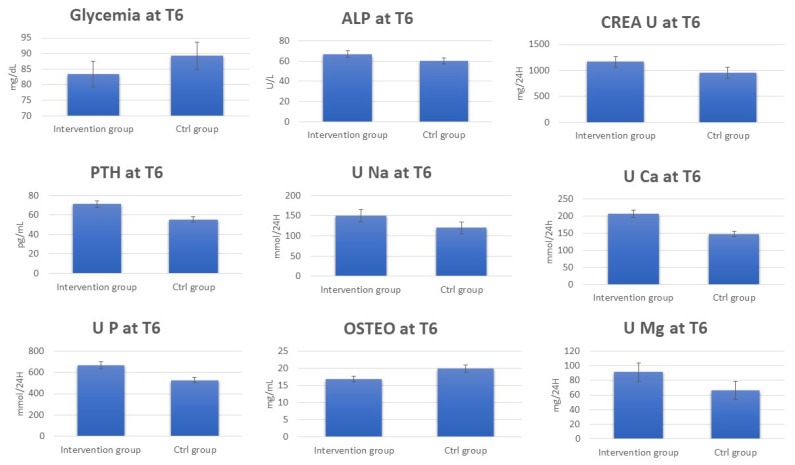
Change in blood and urine concentrations of several important biomarkers of women subjected to 6 months of Ca–bicarbonate water diet–therapy treatment and women subjected to CO_2_-supplemented oligomineral water consumption for 6 months.

**Figure 2 metabolites-13-01109-f002:**
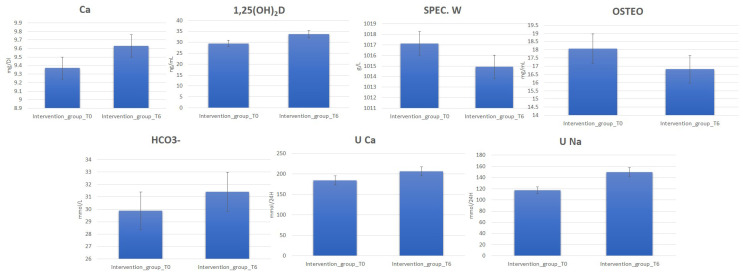
Change in blood and urine concentrations of several important biomarkers of women subjected to 6 months of Ca–bicarbonate-rich water diet–therapy treatment. Clinical parameters are collected at T0 and T6 (*p*-value < 0.05).

**Figure 3 metabolites-13-01109-f003:**
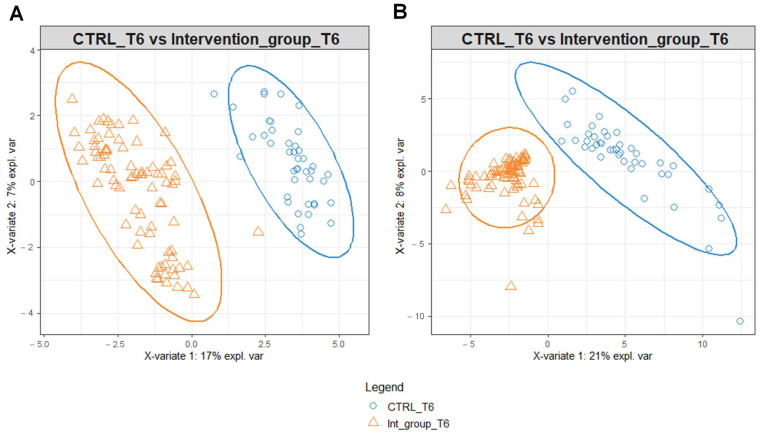
PLS-DA score plot for 1H-NMR collected in 1D-NOESY spectra related to the serum (**A**) and urine (**B**) of the intervention and control group at T6.

**Figure 4 metabolites-13-01109-f004:**
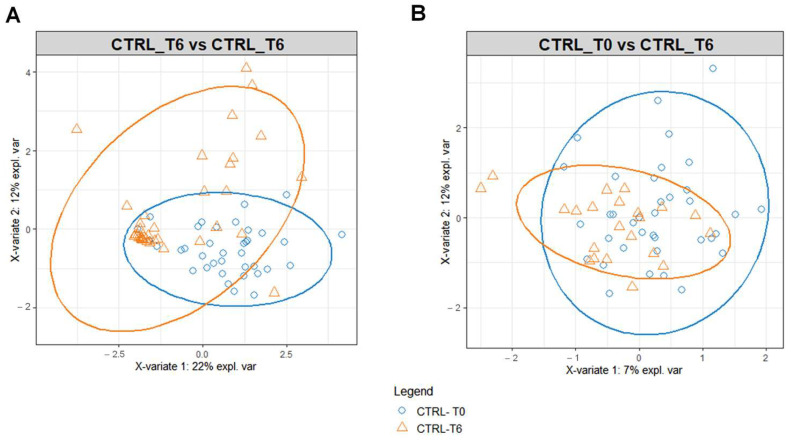
PLS-DA score plot for ^1^H-NMR collected in 1D-NOESY spectra related to the serum (**A**) and urine (**B**) of the control at T0 (blue circles) and control group at T6 (orange triangles).

**Figure 5 metabolites-13-01109-f005:**
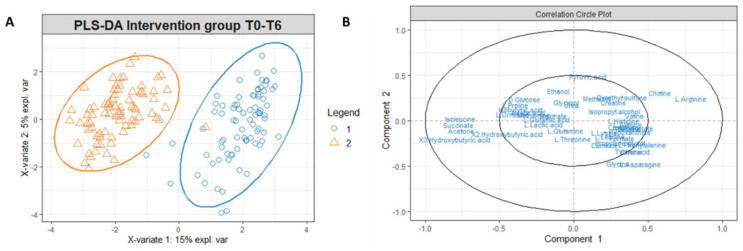
(**A**) PLS-DA score plot for ^1^H-NMR collected in 1D-NOESY spectra related to the serum of the intervention group at T0 (blue circles) and intervention group at T6 (orange triangles). (**B**) Correlation circle plot applied to Component 1 and Component 2. Metabolites that have a greater distance from the origin of the axes represent the variables that most affect clustering. The proximity of the metabolites shows the correlation between the metabolites.

**Figure 6 metabolites-13-01109-f006:**
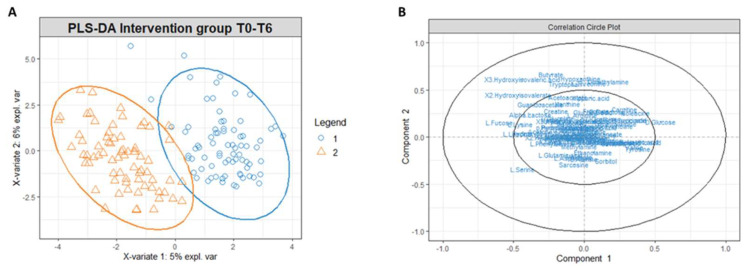
(**A**) PLS-DA score plot for 1H NMR collected in 1D ^1^H NOESY spectra related to the urine of the intervention group at t0 (blue circles) and intervention group at t6 (orange triangles). (**B**) Correlation circle plot applied to Component 1 and Component 2. Metabolites that have a greater distance from the origin of the axes represent the variables that most affect clustering. The proximity of the metabolites shows the correlation between the metabolites.

**Figure 7 metabolites-13-01109-f007:**
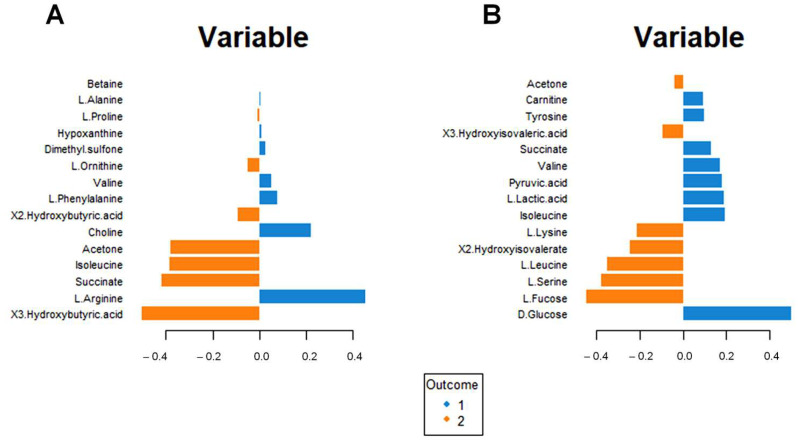
Loading plot related to the matrix of serum (**A**) and urinary (**B**) metabolites. The variables responsible for metabolomic profile differences are ordered according to values of increasing importance from bottom to top. Colors indicate the cluster where the median is maximum for each metabolite (blue: Intervention group T0; orange: Intervention group T6). The metabolite corresponding to an orange band has high concentrations in the intervention group at T6, while the metabolite corresponding to a blue band has higher concentrations in the intervention group at T0.

**Figure 8 metabolites-13-01109-f008:**
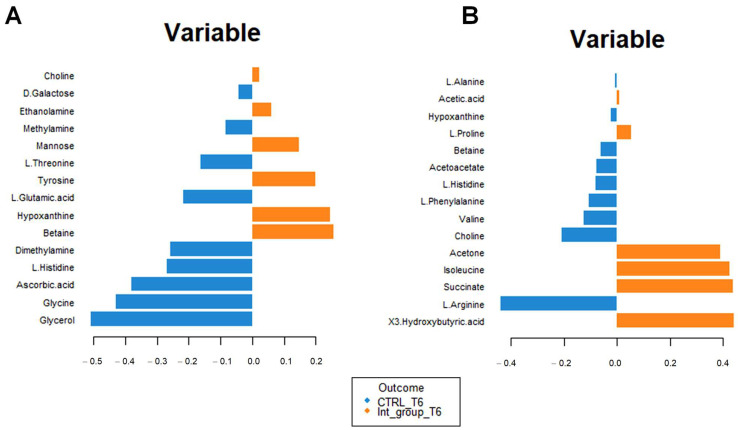
Loading plot related to the matrix of serum (**A**) and urinary (**B**) metabolites. The variables are ordered according to values of increasing importance from bottom to top. Colors indicate the cluster where the median is maximum for each metabolite (blue: control group T6; orange: intervention group T6). The metabolite corresponding to an orange band has high concentrations in the intervention group at T6, while the metabolite corresponding to a blue band has higher.

**Figure 9 metabolites-13-01109-f009:**
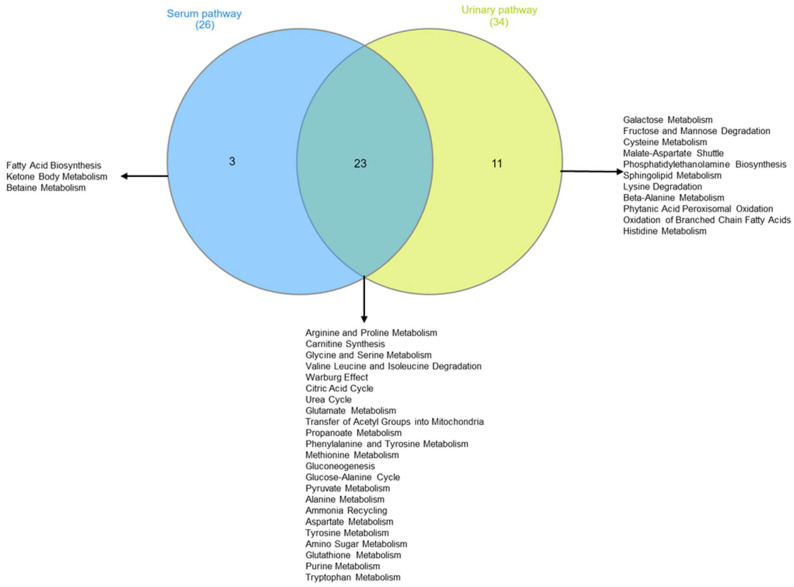
Two-sets Venn diagram intersecting metabolomic pathway identified by pathway enrichment analysis of intervention group serum and urine.

**Table 1 metabolites-13-01109-t001:** Demographics and clinical information of the participants.

Demographics and Clinical Information	Intervention Group	Control Group
Age (years) (average ± SD)	47.04 ± 4.63	47.00 ± 3.81
Height (cm) (average ± SD)	163.67 ± 6.20	161.19 ± 6.31
Weight (Kg) (average ± SD)	72.56 ± 11.52	66.99 ± 9.18
Body Mass Index (kg/m^2^) (average ± SD)	27.30 ± 4.20	26.04 ± 3.43
Education (%)		
Primary	26.28%	28.57%
Secondary	43.47%	42.85%
University	30.25%	28.58%
Smoking (%)	21.73%	33.33%
Physical activity (%)	34.78%	38.09%

**Table 2 metabolites-13-01109-t002:** Bicarbonate–calcic composition.

Elements	Intervention Group	Control Group
Calcium	305	86.2
Magnesium	13.1	12.0
Sodium	5.1	3.4
Potassium	1.9	1.0
Bicarbonates	930	310
Chlorides	10.2	5.2
Nitrates	5.1	3.1
Fluorides	0.3	0.1
Silica	9.2	4.0

**Table 3 metabolites-13-01109-t003:** Metabolic pathways corresponding to serum and urinary metabolomic profile of intervention group. The pathways are classified according to *p*-values, Holm adjustment, and FDR values. The pathways are selected based on Hits > 2, Raw *p* < 0.05, and Holm adjust, FDR < 1.

**Serum_Pathway**	**Hits**	**Raw *p***	**Holm *p***	**FDR**
Arginine and Proline Metabolism	8	1.95 × 10^−71^	1.33 × 10^−69^	6.74 × 10^−70^
Carnitine Synthesis	4	1.58 × 10^−55^	1.06 × 10^−53^	3.63 × 10^−54^
Glycine and Serine Metabolism	10	9.23 × 10^−52^	6.09 × 10^−50^	1.59 × 10^−50^
Fatty Acid Biosynthesis	3	4.22 × 10^−26^	2.74 × 10^−24^	5.82 × 10^−25^
Ketone Body Metabolism	3	5.68 × 10^−22^	3.58 × 10^−20^	5.59 × 10^−21^
Valine Leucine and Isoleucine Degradation	6	3.10 × 10^−17^	1.92 × 10^−15^	2.67 × 10^−16^
Warburg Effect	7	2.36 × 10^−12^	1.44 × 10^−10^	1.81 × 10^−11^
Citric Acid Cycle	3	8.33 × 10^−12^	5.00 × 10^−10^	4.98 × 10^−11^
Betaine Metabolism	3	6.87 × 10^−11^	3.92 × 10^−9^	3.65 × 10^−10^
Urea Cycle	7	8.80 × 10^−11^	4.93 × 10^−9^	4.34 × 10^−10^
Glutamate Metabolism	6	3.36 × 10^−10^	1.85 × 10^−8^	1.49 × 10^−9^
Transfer of Acetyl Groups into Mitochondria	3	3.83 × 10^−10^	2.07 × 10^−8^	1.49 × 10^−9^
Propanoate Metabolism	3	1.89 × 10^−9^	9.24 × 10^−8^	6.19 × 10^−9^
Phenylalanine and Tyrosine Metabolism	4	1.99 × 10^−9^	9.56 × 10^−8^	6.25 × 10^−9^
Methionine Metabolism	4	2.58 × 10^−9^	1.19 × 10^−7^	7.42 × 10^−9^
Gluconeogenesis	3	4.86 × 10^−9^	2.19 × 10^−7^	1.34 × 10^−8^
Glucose–Alanine Cycle	4	5.85 × 10^−9^	2.57 × 10^−7^	1.55 × 10^−8^
Pyruvate Metabolism	4	3.31 × 10^−7^	1.32 × 10^−5^	7.61 × 10^−7^
Alanine Metabolism	4	4.13 × 10^−7^	1.61 × 10^−5^	9.18 × 10^−7^
Ammonia Recycling	6	7.42 × 10^−7^	2.75 × 10^−5^	1.55 × 10^−6^
Aspartate Metabolism	5	1.71 × 10^−6^	6.17 × 10^−5^	3.48 × 10^−6^
Tyrosine Metabolism	3	3.34 × 10^−6^	0.000107	6.07 × 10^−6^
Amino Sugar Metabolism	4	5.58 × 10^−6^	0.000173	9.87 × 10^−6^
Glutathione Metabolism	3	7.06 × 10^−5^	0.001766	0.000108
Purine Metabolism	4	0.000228	0.00524	0.000334
Tryptophan Metabolism	3	0.000626	0.013152	0.000882
Urinary_Pathway	Hits	Raw *p*	Holm *p*	FDR
Amino Sugar Metabolism	5	7.64 × 10^−30^	6.03 × 10^−28^	6.03 × 10^−28^
Galactose Metabolism	6	1.07 × 10^−25^	8.33 × 10^−24^	4.22 × 10^−24^
Fructose and Mannose Degradation	4	2.90 × 10^−25^	2.23 × 10^−23^	7.00 × 10^−24^
Glycine and Serine Metabolism	16	3.54 × 10^−25^	2.69 × 10^−23^	7.00 × 10^−24^
Urea Cycle	11	3.73 × 10^−17^	2.76 × 10^−15^	4.92 × 10^−16^
Ammonia Recycling	9	1.10 × 10^−16^	8.04 × 10^−15^	1.24 × 10^−15^
Arginine and Proline Metabolism	13	4.19 × 10^−16^	3.02 × 10^−14^	4.14 × 10^−15^
Glutamate Metabolism	10	1.17 × 10^−15^	8.28 × 10^−14^	1.02 × 10^−14^
Aspartate Metabolism	9	4.30 × 10^−14^	3.01 × 10^−12^	3.39 × 10^−13^
Alanine Metabolism	6	2.64 × 10^−13^	1.80 × 10^−11^	1.74 × 10^−12^
Cysteine Metabolism	4	6.98 × 10^−12^	4.68 × 10^−10^	4.24 × 10^−11^
Valine Leucine and Isoleucine Degradation	10	7.53 × 10^−12^	4.97 × 10^−10^	4.25 × 10^−11^
Propanoate Metabolism	5	8.90 × 10^−11^	5.52 × 10^−9^	3.91 × 10^−10^
Methionine Metabolism	8	9.57 × 10^−11^	5.84 × 10^−9^	3.98 × 10^−10^
Purine Metabolism	7	1.27 × 10^−10^	7.64 × 10^−9^	5.03 × 10^−10^
Glutathione Metabolism	5	4.09 × 10^−9^	2.41 × 10^−7^	1.54 × 10^−8^
Carnitine Synthesis	6	1.11 × 10^−8^	6.42 × 10^−7^	3.98 × 10^−8^
Warburg Effect	10	1.71 × 10^−8^	9.73 × 10^−7^	5.86 × 10^−8^
Citric Acid Cycle	6	3.46 × 10^−8^	1.87 × 10^−6^	1.04 × 10^−7^
Glucose–Alanine Cycle	5	3.55 × 10^−8^	1.88 × 10^−6^	1.04 × 10^−7^
Gluconeogenesis	5	1.71 × 10^−7^	8.88 × 10^−6^	4.82 × 10^−7^
Malate–Aspartate Shuttle	4	1.10 × 10^−6^	5.40 × 10^−5^	2.81 × 10^−6^
Phosphatidylethanolamine Biosynthesis	3	1.41 × 10^−6^	6.77 × 10^−5^	3.48 × 10^−6^
Sphingolipid Metabolism	4	1.51 × 10^−6^	7.10 × 10^−5^	3.62 × 10^−6^
Phenylalanine and Tyrosine Metabolism	6	1.68 × 10^−6^	7.72 × 10−5	3.90 × 10^−6^
Tyrosine Metabolism	10	3.59 × 10^−6^	0.000162	8.11 × 10^−6^
Lysine Degradation	3	4.10 × 10^−6^	0.000181	9.00 × 10^−6^
Tryptophan Metabolism	5	5.89 × 10^−6^	0.000253	1.26 × 10^−5^
Beta−Alanine Metabolism	6	2.18 × 10^−5^	0.000915	4.53 × 10^−5^
Phytanic Acid Peroxisomal Oxidation	3	2.71 × 10^−5^	0.001111	5.49 × 10^−5^
Transfer of Acetyl Groups into Mitochondria	4	3.15 × 10^−5^	0.00126	6.22 × 10^−5^
Oxidation of Branched Chain Fatty Acids	4	8.88 × 10^−5^	0.003462	0.000171
Pyruvate Metabolism	4	0.00036	0.012612	0.000633
Histidine Metabolism	4	0.018303	0.45758	0.02629

## Data Availability

The data is available within the article.

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
