# Peer review of "Supplementing Low-Sodium Bicarbonate–Calcic (Lete)® Water: Effects in Women on Bone and Systemic Metabolism"

_metabolites, 2023, doi:10.3390/metabo13111109_

Round 1

Reviewer 1 Report

The paper entitled “Supplementing low-sodium bicarbonate-calcic (Lete)® water: effects in women on bone and systemic metabolism.” is presented.  The study is potentially interesting. The methods used were appropriate and the conclusions were justified. 

There were some possible issues.

Figure 1. Change in blood and urine concentrations of several important biomarkers of women sub-268 jected to 6 months Ca-bicarbonate water dietotherapy treatment and women subjected to CO2-269 added oligomineral water for 6 months. Do you have measurements on serum N-terminal propeptide of type I procollagen (PINP, bone formation) and serum C-terminal telopeptide of type I collagen (CTX),

Many studies have found (for example, PMID: 22465238 ; PMID: 32006260) bone homeostasis is regulated by osteoclasts and osteoblasts together with osteocytes, bone lining cells, osteomacs, and vascular endothelial cells in the bone microenvironment within the basic multicellular unit (BMU). It would be informative to discuss the effect of low-sodium bicarbonate-calcic (Lete)®  on these cell types in the bone microenvironment.  

Author Response

REV 1

The paper entitled “Supplementing low-sodium bicarbonate-calcic (Lete)® water: effects in women on bone and systemic metabolism.” is presented.  The study is potentially interesting. The methods used were appropriate and the conclusions were justified. 

There were some possible issues.

Figure 1. Change in blood and urine concentrations of several important biomarkers of women subjected to 6 months Ca-bicarbonate water dietotherapy treatment and women subjected to CO2 added oligomineral water for 6 months. Do you have measurements on serum N-terminal propeptide of type I procollagen (PINP, bone formation) and serum C-terminal telopeptide of type I collagen (CTX).

We thank the reviewer for his/her kind comment. Being the central objective of the study the evaluation of the change of the metabolomic profile after the intake of calcium bicarbonate water compared to the intake of water added with CO2 we have not evaluated PINP and CXC and we focused on the classic evaluations of hematobiochemical parameters. Although a further deepening of the study more focused on the role of collagen will take into consideration these parameters

Many studies have found (for example, PMID: 22465238 ; PMID: 32006260) bone homeostasis is regulated by osteoclasts and osteoblasts together with osteocytes, bone lining cells, osteomacs, and vascular endothelial cells in the bone microenvironment within the basic multicellular unit (BMU). It would be informative to discuss the effect of low-sodium bicarbonate-calcic (Lete)®  on these cell types in the bone microenvironment.  

We thank the reviewer for his/her kind comment.

Unfortunately, it was not possible to make an histological evaluation of patients’ bone tissues so we couldn’t understand the specific action on osteoblastic and osteoclastic cells.

The rebalancing of metabolomic pathways presupposes greater osteoblastic than osteoclastic activities; similarly to the clinical parameters variations observed. Previous scientific studies have shown that variation of some metabolites such as the increment  3-hydroxybutyrate (Figure 8A) and the upregulation of energy pathways (Table 4) affects the differentiation of osteoblasts supporting our hypothesis.[1-3]

Having regard to the correct comments of the reviewer, we have added what was explained earlier in the discussion section of the manuscript (lines 467-477)

Reviewer 2 Report

In this paper, the authors analyzed the effects of taking a commercial water rich in calcium and low in sodium on both bone metabolism and metabolomics.

The manuscript is clear, the results are interesting and well discussed. However, due to the abundance of data (4 tables, 10 figures, 7 supplementary tables, 5 supplementary figures) the results section is difficult to follow. I suggest the authors try to summarize the results and show what is really important and significant. Also, several inaccuracies and some minor points need to be corrected to improve the quality of the manuscript before publication:

1.           Lines 186-187: The average Ca intake of participants before treatment is a very important parameter that should be clearly reported in the results section.

2.           Lines 189-190: I find it difficult to conclude from Supplementary Table 1 that "a Mediterranean diet emerged."

3.           Supplementary Tables 2-5 should be mentioned in the results section.

4.           Figures 1 and 2 do not show P values.

5.           Lines 310-311: Supplementary Tables 4-5 have nothing to do with metabolomic profiles.

6.           Figures 6B and 7B are not clear.

7.           The results reported in Supplementary Table 4 are unrealistic.

8.           Lines 447-448: show statistical analysis related to "glucose decrease."

Author Response

REV 2

In this paper, the authors analyzed the effects of taking a commercial water rich in calcium and low in sodium on both bone metabolism and metabolomics.

The manuscript is clear, the results are interesting and well discussed. However, due to the abundance of data (4 tables, 10 figures, 7 supplementary tables, 5 supplementary figures) the results section is difficult to follow. I suggest the authors try to summarize the results and show what is really important and significant.

We thank the reviewer for his/her kind comment. We streamlined the results to make them more readable

Also, several inaccuracies and some minor points need to be corrected to improve the quality of the manuscript before publication:

  1. Lines 186-187: The average Ca intake of participants before treatment is a very important parameter that should be clearly reported in the results section.

We thank the reviewer for his/her kind comment; we add this information in the results.

  1. Lines 189-190: I find it difficult to conclude from Supplementary Table 1 that "a Mediterranean diet emerged."

We thank the reviewer for his/her kind comment. We evaluated the eating habits of patients and considering that the diet of all patients was based on regular consumption of olive oil (as the main source of added fat), vegetable foods (cereals, fruits, vegetables, legumes, nuts , and seeds), moderate consumption of fish, seafood and dairy products and low to moderate alcohol consumption (especially red wine)balanced by a relatively limited use of red meat and other meat products. Therefore, it was possible to conclude that this regime was Mediterranean. The eating habits are in fact similar to those described in the literature of the Mediterranean diet.[4-6]

  1. Supplementary Tables 2-5 should be mentioned in the results section.

We thank the reviewer for his/her kind comment; we add this information in the results

  1. Figures 1 and 2 do not show P values.

      We thank the reviewer for his/her kind comment; we add this information in the Supplementary Tables

  1. Lines 310-311: Supplementary Tables 4-5 have nothing to do with metabolomic profiles.

We thank the reviewer for his/her kind comment; we replace “Supplementary Tables 4-5” with Supplementary Table 6-7

  1. Figures 6B and 7B are not clear.

We thank the reviewer for his/her kind comment; we add a more detailed description in the caption.

  1. The results reported in Supplementary Table 4 are unrealistic.

We thank the reviewer for his/her kind comment; unfortunately, there was a transcription error; we corrected the error.

  1. Lines 447-448: show statistical analysis related to "glucose decrease."

We thank the reviewer for his/her kind comment; we add the information in the manuscript

Reviewer 3 Report

In this manuscript, by a complete set of laboratory tests, the authors investigated supplement six months of Lete water (rich in calcium and bicarbonate, and low in sodium) whether to benefit the Ca metabolism and bone health and using NMR-based metabolomic procedures to study potential system-wide metabolic effects.

The authors found that water rich in Ca and poor in sodium intake for six months, in addition to inducing an improvement of the biochemical parameters related to Ca metabolism and bone health, it also could restore mitochondrial energy pathways, promote the application of bone mass and collagen, and reduce the biomarkers of osteoporosis.

 In this manuscript, the authors try to present a large amount of data to the reader. However, in my opinion, this makes the manuscript a bit complicated and lengthy to read. In fact, the manuscript can be more concise, especially the results section, which can be adding more data to supplement file and to make the article clearer and more focused on the authors' findings. For examples, in results part, 3.1 of the manuscript, it is best to display and only display the T0 and T6 biomarkers data that have significantly changed between the control and intervention groups in a concise and clear diagram. In part 3.2 of the manuscript, Figure 3 and figure 4 can be combined into one figure, and Figure 3A could be arranged with Figure 4 A, Figure 3B could be arranged with Figure 4B.  In this way, the changes between control group and intervention groups are clear at a glance.

There are some errors in the tables of supplementary that need to be correct carefully, especially the decimal points.

Write the results section of manuscript concisely to highlight what the authors findings. 

Author Response

inducing an improvement of the biochemical parameters related to Ca metabolism and bone health, it also could restore mitochondrial energy pathways, promote the application of bone mass and collagen, and reduce the biomarkers of osteoporosis.

 In this manuscript, the authors try to present a large amount of data to the reader. However, in my opinion, this makes the manuscript a bit complicated and lengthy to read. In fact, the manuscript can be more concise, especially the results section, which can be adding more data to supplement file and to make the article clearer and more focused on the authors' findings. For examples, in results part, 3.1 of the manuscript, it is best to display and only display the T0 and T6 biomarkers data that have significantly changed between the control and intervention groups in a concise and clear diagram. In part 3.2 of the manuscript, Figure 3 and figure 4 can be combined into one figure, and Figure 3A could be arranged with Figure 4 A, Figure 3B could be arranged with Figure 4B.  In this way, the changes between control group and intervention groups are clear at a glance.

There are some errors in the tables of supplementary that need to be correct carefully, especially the decimal points.

We thank the reviewer for his/her kind comment we modified the manuscript

Round 2

Reviewer 2 Report

None

Reviewer 3 Report

Thanks for the author's reply. This version is much better than the first one, and no more comment.